# Leveraging Vector-Based Gene Disruptions to Enhance CAR T-Cell Effectiveness

**DOI:** 10.3390/cancers17030383

**Published:** 2025-01-24

**Authors:** Beatriz C. Oliveira, Saaurav Bari, J. Joseph Melenhorst

**Affiliations:** Cell Therapy & Immuno-Engineering Program, Center for Immunotherapy and Precision Immuno-Oncology, Lerner College of Medicine, Cleveland Clinic, Cleveland, OH 44016, USA; coutinb@ccf.org (B.C.O.); baris@ccf.org (S.B.)

**Keywords:** CAR T cell, vector integration, genomic disruptions, hematological malignancies

## Abstract

In chimeric antigen receptor (CAR) T-cell therapies, T cells from patients are reprogrammed to target their tumor cells. This is done using viral vectors to insert the CAR gene into the T-cell genome. While many heavily pretreated patients show response and CAR T-cell expansion, others do not. Our findings suggest that the CAR genes’ insertion location can influence clinical effectiveness by altering cellular genes. However, the biology of many insertion sites is still not fully understood. This review presents current insights into the link between vector integration and CAR T-cell efficacy.

## 1. Introduction

Chimeric antigen receptor (CAR)-engineered T cells have revolutionized cancer immunotherapy as they are able to provide significant anti-tumor and long-term responses in patients with hematological malignancies [1,2,3,4,5,6,7]. In this approach, the patient goes through a blood separation process known as apheresis, where their peripheral blood mononuclear cells (PBMCs) are collected. The T cells are further isolated, activated, and engineered to stably express a tumor-targeting chimeric receptor that recognizes a tumor-associated cell surface molecule, i.e., CD19 on B-cell malignancies [8,9]. After expansion, these genetically modified cells are reinfused into the patient, where they will effectively target and kill the tumor cells (Figure 1). Viral vectors are commonly used to deliver the CAR predominantly in or near actively transcribed genes and, by virtue of this insertion event, potentially disrupt the structure and/or function of the affected genomic site. In gene therapies, gammaretroviruses have been used due to their ability to carry large transgenes and because of the facilitated production using stable cell lines, enabling high transduction rates into target cells [10]. However, with their increased use, safety concerns started to emerge, as they tend to integrate their transgenes near oncogene promoters, posing a potential risk of insertional mutagenesis and transformation [11]. Considering those events, lentiviral vectors became an attractive and safer alternative. These vectors share structural similarities with gamma retroviruses, as both allow for the replacement of essential viral genes with a transgene of interest, enabling a stable integration in host cells. Lentiviral vectors employ a split-genome design, wherein viral genes and regulatory elements are separated to minimize the risk of recombination. Importantly, lentiviral vectors exhibit a distinct integration profile, favoring insertion into actively transcribed genes, in contrast to gamma retroviral vectors, which preferentially integrate near transcription start sites and CpG islands [12,13]. The common design of the CAR consists of three domains: (a) the extracellular; (b) the transmembrane; and (c) the intracellular. The extracellular domain is composed by the single chain variable fragment (scFv), an engineered polypeptide derived from both heavy and light chains of the native antibody [9,14]. The scFv specifically binds to the target, and that interaction culminates in T-cell activation, triggered by the intracellular domain (CD3ζ) derived from the naïve T-cell receptor (TCR) signaling complex. Different elements on the intracellular component of the CAR can generate potent responses. First-generation CARs are solely composed of the CD3ζ signaling domain, which only provides signal 1 (antigen-specificity signal), therefore not leading to sufficient activation. The second and third CAR generations contain at least one costimulatory domain to allow for the full activation of CAR T cells with potent antigen-specific cytotoxicity. The commonly used costimulatory domains are derived from CD28 and CD137 (4-1BB), with the latter providing superior in vivo persistence of CAR T cells [15]. Despite great success, in some cases, especially in the context of chronic lymphocytic leukemia (CLL), where T cell-intrinsic defects can lead to a poor response, only about 30% will achieve durable remissions [16,17,18,19,20,21,22]. The mechanisms by which the cancer cells negatively impact CAR T potency, on the other hand, are still not fully understood.

Several factors play a role in determining the resistance mechanisms involved with treatment failure, as they can be primary (lack of initial response) or acquired (relapsed response) [23]. Nonetheless, while a few biological processes have been implicated with acquired resistance and antigen loss [24,25,26,27], there is still a gap in our knowledge regarding why some patients do not respond to CAR T-cell therapy. One hypothesis suggests that transcriptional programs regulating the fate of CAR T cells, along with their clonal composition and kinetics post-infusion, have a significant impact, potentially providing insights into disease prognosis [28]. It has indeed been reported that an early memory population, which displays increased self-renewal ability, followed by the functional persistence of CAR T cells, is required to achieve sustained tumor control [7,28,29]. This profile was similarly observed in two-decade-long CLL remission patients who were followed up by our group [30]. Several factors, such as the CAR design, the manufacturing conditions (including activation reagents and T-cell isolation methods), the quality of T cells in the starting product, the population of the infused cells, and the tumor microenvironment can also highly influence the generation of memory cells [6,15]. Understanding all these factors, and, importantly, the repertoires contained within the CAR T-cell product is crucial for optimizing CAR T-cell therapies, as they can play pivotal roles in achieving sustained tumor control and long-term remission. Two major ways of deciphering these repertoires include the sequencing of the T-cell receptor (TCR) and the lentiviral vector integration sites (LVIS), which not only provide insights into clonal tracking but can also be informative for understanding how vector insertions contribute to different CAR T-cell responses [31,32]. In this manuscript, we review how LVIS-based gene disruptions account for important functional consequences and how they can provide us with valuable predictions of treatment outcomes and CAR T-cell efficacy. By identifying various pathways, mechanisms, and genes associated with the therapeutic proliferation of CAR T cells, LVIS analyses have the potential to offer important insights for enhancing T-cell function in immunotherapies. Collectively, they could provide alternatives for engineering T cells to achieve optimal effectiveness.

## 2. Viral Integration in Human Gene Therapies

The ability of oncoretroviral and lentiviral vectors to stably insert the payload into the host’s genome has been a strong driver of research on safety and efficacy of this mode of gene correction [32,33,34,35,36]. The steps included in this process are summarized in Figure 2. Once the virus enters the cells and has their RNA converted into DNA by the reverse transcriptase, the integration process starts. An enzyme, namely integrase, cleaves the long terminal repeats (LTR) located at edge of the proviral DNA and incorporates the transgene into the host’s chromatin. DNA-repair enzymes are responsible for sealing gaps and finishing the integration process [13].

The identification of clonally selected T cells harboring the viral genome in specific sites, as observed in HIV-infected T cells and patients with adverse events following gene therapy, has generated a strong impetus to understand the viral genome insertion sites and how they relate to the clonal dynamics of the infected cells. This impact occurs through the disruption of native gene expression or regulatory elements, potentially altering cell differentiation or activation processes [37,38]. The first reports of integration site analysis in oncoretroviruses revealed a preferential integration of the viral genome in actively transcribed genes [13]. It was then confirmed that the integration pattern was occurring within certain genomic regions, i.e., near promoter regions in oncogenes including those actively transcribed [39]. Studies employing techniques such as ligation-mediated PCR, which facilitate a more in-depth analysis, have challenged the previously held belief about the nature of vector integration sites and their viral populations [37,40]. These studies suggest that, unlike what was previously believed, the nature of the insertion sites might be more dynamic, with nonlinear accumulation of sequence diversity during the course of infection [41,42].

Safety issues associated with gene therapies aimed at correcting defective native genes concerned the disruption of cellular oncogenes in the transduced cells, resulting in oncogenic growth of the modified cells [43]. X-linked severe combined immunodeficiency (X-SCID), a genetic disorder solely affecting male children, results from mutations in the IL-2 receptor common gamma chain (CD130), resulting in uncontrolled, persistent infections [44]. A gene therapy addressing this deficiency restored CD130 expression by using gammaretroviral vectors, being successful in nine out of ten patients [44]. However, four out of the nine developed T-cell leukemia several years later, and that was likely due to vector integration into two proto-oncogenes, LMO2 and BMI1, both known for their roles in leukemic cell development [35]. A second example is the successful treatment of β-thalassemia, which was achieved with a lentiviral vector encoding a β-globin flanked by insulator elements, but concerns regarding long-term safety arose when clonal expansion occurred after integration in the HMGA2 locus [45]. Therefore, it was uncertain whether clonal proliferation of cells carrying vectors integrated near growth control genes was inevitable or whether safety concerns are limited to specific cell types.

A pivotal insight into the impact of gene correction in mature T cells was provided during the first SCID-X1 gene therapy trial [46]. It was observed that retroviral vector integration sites were not random but instead showed significant clustering in multiple common integration sites. Notably, these sites were often located in growth-regulating genes, suggesting a preference for integration into active gene loci. Integration in vector-targeted genes or their regulatory regions likely confers a clonal selection advantage in subsequent cell generations compared to sites that are rarely affected. This pattern of preferential vector insertion was later corroborated by Deichmann et al. [39], who identified four distinct mechanisms by which retroviruses integrate into the genome and how they were linked with cell growth/proliferation. Therefore, long-term surveillance of the integration sites is required to predict possible adverse effects.

## 3. Vector-Based Disruptions Correlated with Clonal Expansion

In one of the largest studies of its kind, using a retrospective cohort, Nobles et al. [31] screened lentiviral vector integration sites (LVIS) in 40 patients who underwent autologous anti-CD19 CAR T-cell therapies. Their comprehensive analysis identified 145,600 unique integrations mapped to the human genome based on 184 samples collected both before and after infusion from these 40 patients. Because single genomic insertions were favored in the CAR T-cell manufacturing process by targeting transduction efficiencies of 20% or less, most T cells carried only single vector insertions. This study therefore offered insight into the clonal dynamics of CAR T cells, as well as the potential impact of potential mutagenesis of affected genes. It identified cell signaling and epigenetic regulators as commonly affected genomic regions in responding patients, suggesting that the insertion into these sites contributed to the in vivo expansion and clinical efficacy of the inserted sites. Additionally, while some genes carried the vector insertion in multiple patients, the same site was never affected [31]. To investigate the impact of such events long-term, as the follow-up in that specific cohort varied from 28 days to 5 years, additional retrospective studies are needed in patients who share the same gene but with different site integrations. Further, the characterization of LVIS in the ultralong persisting CAR T cells of two patients who were cured of their leukemia identified a significant clonal selection and persistence of CAR T cells over the course of their follow-up but with a strong CD4 dominance in the second half of their decade-long remission [30]. It is expected that transduction of cells at higher multiplicity of infection (MOI) has a higher likelihood in generating cells carrying integrations in multiple genes, as observed by vector copy number calculations using digital droplet PCR (ddPCR) [47]. The impact of such integration sites obviously depends on the precise integration site and the combination(s) of gene disruptions. Multiple insertion sites could lead to dramatically augmented cell growth, but such findings have yet to be reported and assessed for safety in CAR T-cell trials.

The scenario might be different in solid tumors. It was described in a longitudinal study on glioblastoma patients who underwent gene therapy [48] that, although multiple retroviral integration sites were detected, the number of clones steadily declined over time after treatment was discontinued. The patients had normal cytogenetics and showed no evidence of leukemia or myelodysplastic syndrome. However, as reported by Howe et al. [11], where a patient has developed acute lymphoblastic leukemia (ALL) after successful gene therapy against X-SCID, it is important to consider potential factors that might contribute to adverse events caused by insertional mutagenesis, such as viral infections and/or genetic predisposing factors. We emphasize that, in this trial, a gamma-retroviral vector was used, and new generations of lentiviral vectors developed more recently address safety concerns related to vector-mediated malignant transformation. In the context of CAR T-cell therapy, our group reported the case of a CLL patient who had a clonal expansion of a single CD8 T cell containing a single-allele mutation on the TET2 gene, caused by lentiviral integration, leading to their complete remission [49]. CAR T-cells from this patient, who still had the other allele intact, although hypofunctional, were characterized by a predominant central memory phenotype and enhanced effector function. The complete (biallelic) ablation of TET2 expression in CAR T-cells, on the other hand, resulted in the uncontrolled proliferation of T cells that had lost their effector function [50]. It is important to mention that the TET2 gene has an important role in hematopoiesis, and its complete disruption might potentially lead to hematological malignancies [51].

Similar to the TET2 case, a 28-year-old patient who presented with relapsed pre-B-cell ALL after allogenic hematopoietic stem cell transplant (HSCT) and blinatumomab (a bispecific antibody targeting CD19 and promoting in vivo engagement of T cells—BiTE) treatment was part of a CD22-targeted CAR T-cell trial designed for ALL [52]. TCR sequencing and LVIS analysis of the infusion product indicated a lentiviral integration on the Casitas B-lineage lymphoma (CBL) gene [52]. CBL-b is an E3 ubiquitin ligase and a negative regulator of T-cell activation, influencing CD28 dependence [53,54]. Clinical data reported on the CBL case [52] indicate that the first peak of CAR T-cell expansion exhibited a predominantly polyclonal profile. However, during a second peak of CAR T-cell expansion, one clone harboring an insertion of the CAR vector in the CBL gene comprised around half of the total white blood cells. This particular clone was not identifiable in the infused CAR T product. However, it emerged at a low level (2.6%) as early as day 7 post-infusion during the initial CAR T-cell expansion and remained detectable, albeit below 1%, on days 15 and 30 after infusion in the patient [52]. This single clone underwent a significant expansion from day 51 to day 53 post-infusion, notably without any cytokine release syndrome (CRS) symptoms in the patient [52]. The reason why this second expansion happened was unclear, but it can possibly be partially attributed to the integration into the CBL gene, as the authors discussed the possibilities that: (a) antigen stimulation through the endogenous TCR was unlikely due to negative tests for Epstein–Barr virus or cytomegalovirus infection; or (b) the clone expanded through the CAR due to residual lymphomatous disease [52]. In this case, unlike in TET2, this patient had no mutations on either allele of the CBL gene. Therefore, the lentiviral integration did not lead to a complete gene loss of function. LVIS analysis revealed that the integration occurred in the second intron of one CBL allele [52]. A study in mice where T cells had a single dysfunctional CBL-b allele resulted in no significantly enhanced activity against tumor cells [55], but complete ablation of this gene led to enhanced antitumoral response, and after 1 year follow-up, these mice were still able to respond. Shah et al. [52] hypothesized that the single lentiviral integration yielded either a dominant–negative or interference effect on the function of the CBL/CBL-b gene that led to hyperproliferation of the CD22 CAR T cells, followed by the complete remission of the patient. Regarding both the TET2 and CBL cases, the CAR T cells underwent a delayed proliferation in the presence of residual disease, indicating the possibility that the clones could have a lower activation threshold. It is thus reasonable to hypothesize that the degree of gene disruption can significantly impact cell function and persistence and that subtle changes in gene expression can have significant biological consequences. These examples highlight the importance of understanding the impact of gene disruptions on different cell types and networks, as these factors can have dramatic implications, especially for the efficacy and safety of T cell-based therapies.

However, the claim that such vector integrations are the critical drivers of clonal kinetics of CAR T cells and play a role in therapy outcomes has since been refuted. Sheih et al. [28] examined a subgroup of adult patients (n = 10) who were either affected by relapsed and refractory B-cell ALL, non-Hodgkin lymphoma (NHL), or CLL. These patients were treated with CAR19 T cells containing the 4-1BB intracellular costimulatory domain produced from a combination of bulk CD4+ T cells and CD8+ central memory-enriched (TCM) cells administered in a 1:1 ratio of CD4s:CD8s in a phase 1 clinical trial. The LVIS of CD8+ CAR T cells isolated from the infusion products and from peripheral blood were analyzed on days 7–14 and after peak expansion on days 26–30 in three ALL (n = 3) and four NHL patients [28]. Across all patients and samples, they identified a total of 55,382 unique integration events, with the majority (82.6%) being located within genes, particularly in introns rather than exons, which is in agreement with our findings [31]. They have also identified two categories of integration sites that showed a significant increase or decrease in relative abundance by at least 5-fold between the infusion product and post-infusion samples. In an attempt to observe similarities with the unique TET2 case [49], Sheih et al. [28] reported only 12 integration sites in six patients within the TET2 gene, none of which ranked among the top 20 most abundant sites in any patient or sample. Therefore, the integrations within the TET2 gene were not a prevalent driver of clonal expansion in their study. These findings indicate that, in the cohort studied by Sheih et al. [28], a single vector integration site was unlikely to be the primary driver of clonal kinetics and proliferation. Interestingly, they observed that increasing clonotypes were enriched in integration sites located within genes associated with lymphocyte activation, suggesting a correlation between robust anti-CD3/CD28-mediated T-cell activation before lentiviral transduction and subsequent clonal proliferation [28].

Another similar study [7] analyzed the LVIS at early (1 month) and later (>3 years) time points in a clinical trial (CARPALL—NCT02443831) involving refractory and relapsed pediatric B-cell ALL patients treated with a novel low-affinity CAR19 T-cell product. They demonstrated that their scFV has a significantly lower affinity for CD19 than the CD19-FMC63 scFV, at approximately 40-fold [56]. Their study revealed a polyclonal integration site profile of CAR T cells in the infusion product, with remaining clonal diversity shortly after and in the long term, with no indication of dominant clones [7]. Different clones exhibited varied expansion and contraction patterns over time, with distinct clones contributing to early expansion and sustained persistence. Notably, the study identified and monitored 12 clones in patients with integrations in the TET2 gene. However, none of these clones demonstrated signs of significant clonal dominance and expansion during the initial response or in long-lasting CAR T cells [7]. It is important to note that the lentiviral integration described by Fraietta et al. [49] in the TET2 gene involved a rare occurrence of a hypomorphic mutation in the second TET2 allele, which is unlikely to have occurred in the patients analyzed in this study [7]. Therefore, it is important to note that the degree, location, and type of integration (mono vs. biallelic disruption) plays a decisive role in whether the clonal expansion of these disrupted cells takes place or not.

## 4. Integration Site Analysis Strategies in CAR T-Cell Therapies

The safety profile of retroviral or lentiviral vectors used in CAR T-cell manufacturing has dramatically improved over the past decades [32,40,57,58]. As discussed above, these platforms can sometimes generate unpredictable outcomes, such as malignant transformation or leukemogenesis, and significant research has been performed regarding their safety [59,60,61,62], which is considered relatively high. RNA sequencing performed in a prior study [16] identified the gene expression of the top 500 genes that were active in pre-infusion products of all types of response groups of patients with ALL and CLL, and the frequency of lentiviral integration in these genes was compared among them. Patients who achieved complete response had a high representation of genes involved in early-memory T-cell differentiation and IL-6/STAT3 response, and the genes presented in non-responders were associated with effector T-cell differentiation, exhaustion, glycolysis, and apoptosis. The results obtained in this study allowed us to gain insights into how insertional mutagenesis can influence treatment outcomes [16]. Evaluating a bigger cohort of ALL and CLL patients, we observed not only that it was possible to associate insertion sites with complete response (CR) but that several other gene disruptions were linked with patients who had partial response with transformed disease (PRtd—caused by loss of the CD19 expression in leukemic cells) or no response (NR) [31].

One of the main techniques used to identify CAR insertion sites is by ligation-mediated (LM) PCR [63]. In this approach, the extracted genomic DNA is fragmented through sonication or restriction enzymes, then a linker is ligated to the DNA fragments, serving as a binding site for PCR primers. After ligation, the DNA is subjected to PCR using primers that are specific to the known linkers and to the lentiviral vector. To increase specificity, nested PCR is usually performed using an internal primer from the first PCR product and the lentiviral vector [63]. The final product is then sent for sequencing and analysis to determine the exact integration site in the host genome. Although being the most frequently used techniques in the clinic, linear amplification (LAM)- and LM-PCR have been limited because of technical bias, long protocols, and the need for a more elaborate infrastructure and a relatively high DNA input [64]. To overcome these issues, other techniques were developed and are currently available. A recently developed methodology, namely DIStinct-seq [65], which stands for “detection of the integration sites in a time-efficient manner, quantifying clonal size using tagmentation sequencing”, has improved the clinical feasibility of IS analysis. Researchers have, such as in previous studies [66,67], used the tagmentation technique, which is mediated by a transposase enzyme that simultaneously fragments DNA and adds adapter sequences [68]. However, the system was adapted with a bead-linked Tn5 transposase, which simplified the process as well as enabled the quantification of integration sites. Generally, integration site (IS) analyses assess clonal abundance by employing a stable vector integration as a marker for the clone. Since all progeny cells originating from the same parent cell share the same IS, the IS count can be used to determine the clone’s size. Therefore, the total number of DNA fragments contributing to an IS, also referred to as raw fragment counts (RFC), serves as a measure to quantify the clones. However, PCR amplification biases, influenced by variations in fragment length or GC content, can affect this measurement [65]. As an alternative approach to overcome these limitations, researchers quantified the IS by counting distinct fragment lengths generated from the random shearing of DNA after removing duplicated PCR fragments, termed deduplicated fragment counts (DFC). Therefore, it was possible to experimentally assess the quantification performance of DIStinct-seq using samples with known IS mixed in specific proportions. Finally, it was reported that a relationship between the IS and clonal behavior existed [65], which, in agreement with our previous discoveries [31,49], could be crucial for the development of safe and efficacious CAR T cells.

Another method for LVIS analysis utilizes the CRISPR-Cas9 system for detection. CRISPR-enhanced viral integration site sequencing, known as “CReVIS-Seq” [69], enriches target DNAs using CRISPR-mediated cleavage. By performing an in vitro fragmentation through the sonication and circularization of genomic DNA through T4-DNA ligase followed by CRISPR-Cas9, it is possible to induce cleavage of those DNAs containing lentiviral LTRs, which are generally incorporated into the host genome together with other lentiviral sequences. This method provides some advantages, such as the possibility to detect lentiviral insertion sites in clonal populations derived from single cells, as well as to identify multiple insertion sites in bulk cell populations. Besides that, different sgRNAs for multiple cleavages can be used to simultaneously acquire the DNA sequence information of multiple target loci in the host genome [69]. This method can also be used to enrich multiple targets, meaning that this could be generalized for genome-wide mapping of integration sites. Importantly, it is also worth mentioning another methodology, known as INSERT-Seq [70], which uses the nanopore technology offered by Oxford Nanopore. Its long read lengths allow for comprehensive coverage, overcoming limitations associated with short reads such as in PCR. INSERT-seq can also identify previously unknown integration sites within a modified genome with a detection limit of 1% while maintaining robustness, speed, and cost effectiveness. Moreover, the portability and accessibility of nanopore sequencing platforms enable rapid on-site analysis, streamlining the integration site profiling process and accelerating the translation of research findings into clinical applications [70]. As these technologies (Figure 3) evolve, their integration into the workflow of CAR T-cell therapy development holds immense promise for enhancing safety assessments and optimizing treatment strategies.

## 5. Future Perspectives

Some groups have been driving their efforts to develop non-viral CAR T cells, including those based on the PiggyBac transposon [67,71], mRNA delivery through nanocarriers [72,73,74], electroporation ex vivo [75,76,77], or in vivo using LNPs [78]. The PiggyBac system employs a transposase enzyme to excise the transposon cassette containing the gene of interest and integrate it into the genome at a new site [67]. Even though it has been described as a strategy which promotes a more stable integration profile, flexibility of payload size, and a simpler manufacturing process, it might pose safety concerns regarding off-target effects and malignant transformation [71]. However, it is an approach worth pursuing and one that can lead to promising results when safety is fully addressed. Regarding methods utilizing mRNA delivery, although attractive because they offer a rapid, low-cost, and transient expression of the CAR [79], the field remains largely unexplored, with very few products advancing for clinical trials.

Other non-viral approaches include base and prime editing, which are advanced genome-editing techniques that can promote a precise, site-specific integration of the CAR into T cells [80]. Unlike traditional CRISPR-Cas9 methods, which introduce double-stranded breaks (DSBs) to insert the targeted genetic material, base and prime editors modify specific DNA nucleotides without creating DSBs, thereby reducing the risk of unintended mutations and chromosomal rearrangements [81]. However, despite their high precision, these approaches are not entirely free from off-target effects. Unintended edits can occur at sites with sequences which are similar to the target, potentially leading to aberrant gene function or expression [82,83]. Additionally, the efficiency and specificity of base and prime editing can vary depending on the genomic context and the sequence(s) being targeted [84]. Therefore, while base and prime editing can offer significant advantages regarding precision of CAR integration, comprehensive analyses of off-target activity and their functional consequences are essential to guarantee both the safety and effectiveness of the treatment.

In viral-based CAR T cells, many strategies have been developed to improve this therapy for refractory and relapsed hematological malignancies. These include but are not limited to the following: combination therapies with monoclonal antibodies, checkpoint inhibitors, or immunomodulatory drugs; dual targeting of the CAR; making allogeneic products; and others [85]. Recently, the Food and Drug Administration (FDA) has started an investigation of cases presenting a secondary T-cell cancer followed by CAR T-cell therapy [86]. Two cases were reported after allogeneic CAR T-cell therapy for lymphoma using the PiggyBac Transposon System (System Biosciences), and the trial was discontinued after that finding [71]. In another report, T-cell lymphoma was diagnosed in a patient 3 months after CAR T-cell therapy, in which the CAR copy numbers detected by qPCR were very low (representing roughly 0.005% of the cells), interpreted to be caused by infiltrating CAR T cells rather than the lymphoma harboring the CAR transgene [87], even though this hypothesis should not be disregarded. Another group reported the case of a patient who developed lymphoma 5 months after CAR T-cell therapy, coupled with a single LVIS within tumor cells shown 9 months after treatment, suggesting that CAR T cells probably contributed to lymphomagenesis [88]. Characterizing genes disrupted by vector integration in expanding or persisting CAR T cells using functional genomics (e.g., CRISPR/Cas9 or CRISPR-a/i) and correlating this data with in vitro and in vivo proliferation, as well as anti-tumor activity, is essential. This approach enables efficient interrogation of multiple alleles per gene and facilitates the evaluation of risks, including the emergence of secondary malignancies. Alternatively, using small molecules that pharmacologically inhibit the corresponding protein targets of CAR vector-disrupted genes (considering those happening within their coding regions), one could potentially mimic the outcome of certain perturbations that occur in patients during CAR insertions. Ultimately, interrogating genes that are likely linked to complete and durable responses in patients can help us to better understand which proteins and/or pathways are involved with these positive outcomes.

## 6. Conclusions

As discussed above, understanding the impact of LVIS on CAR T-cell function is essential for advancing this therapeutic approach. Our group has previously reported how specific gene disruptions can influence treatment outcomes [31,49]. However, the role of certain genes identified through patient integration site analyses remains unexplored in the context of T-cell biology. Gaining this knowledge will provide valuable insights into overcoming treatment failure by improving T-cell expansion and persistence. Consequently, these findings will help drive the development of alternative strategies to safely optimize CAR T-cell engineering.

## Figures and Tables

**Figure 1 cancers-17-00383-f001:**
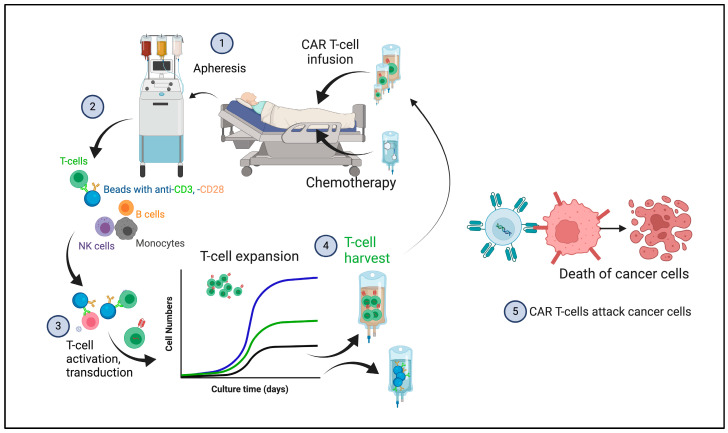
Overview of a general CAR T-cell manufacture process. Patient blood is collected through apheresis (1), and mononuclear cells are enriched using counterflow centrifugation and immunomagnetic bead-based T-cell selection (2). These T-cells are then activated and transduced with a vector encoding the CAR, which randomly inserts the CAR gene into the T-cell genome (3). The T-cells are expanded, and stimulatory beads are removed before infusion (4). After lymphodepleting chemotherapy, CAR T-cells can be infused into the patient, either as a single dose or, as shown here, in escalating doses over consecutive days or weeks; the latter treatment scheme is more often applied in clinical trials, as most FDA-approved products rely on a single dose. Once infused, these CAR T-cells proliferate and target cancer cells (5). (Created in BioRender, https://BioRender.com/d07f736 (accessed on 18 January 2025)).

**Figure 2 cancers-17-00383-f002:**
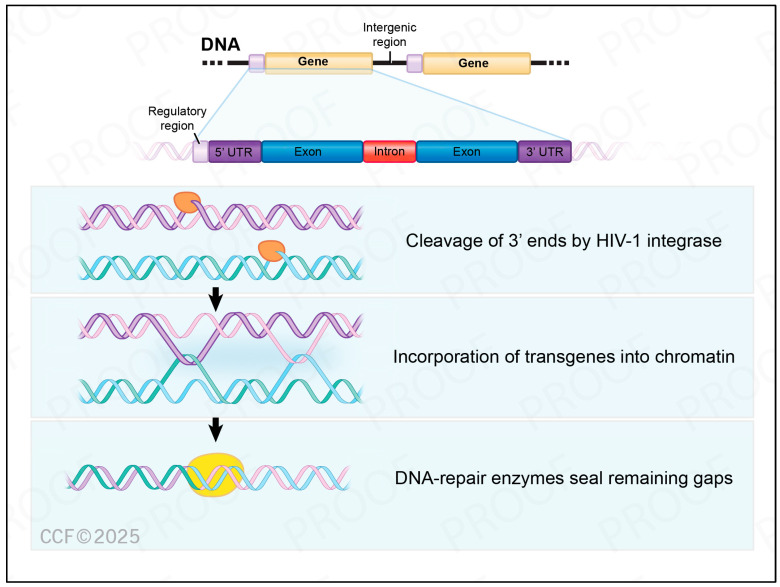
Schematic view of viral integration into the host genome. (Modified from https://BioRender.com/w74a841 (accessed on 18 January 2025)). UTR = untranslated region, LTR = long terminal repeats, located at the edge of the proviral DNA. Retroviruses typically insert within promoters (regulatory region), while lentiviruses have a different pattern of integration, happening frequently among intragenic regions, particularly within introns.

**Figure 3 cancers-17-00383-f003:**
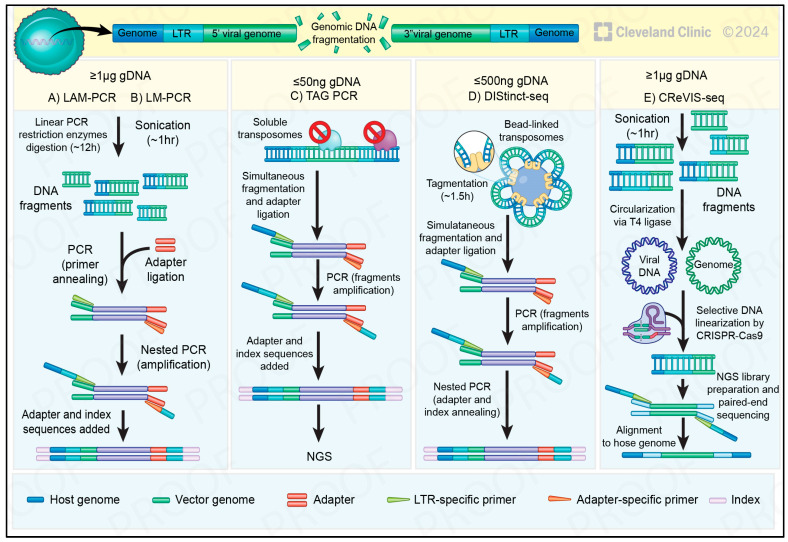
Schematic view of different technologies for integration site (IS) analyses. LAM-PCR = linear-amplification mediated; LM-PCR = ligation-mediated PCR; LTR = long terminal repeats, located at the edge of the proviral DNA; NGS = next-generation sequencing (Illumina platform). (**A**,**B**) Begin with linear PCR, then, to fragment the DNA, restriction enzymes are used for LAM-PCR or sonication for LM-PCR, and at least 1 μg of DNA is required. Next, ligation of adapters to the fragmented DNA is used to create additional sites for PCR primer annealing. Nested PCR is finally performed to specifically amplify host–vector chimeric fragments and incorporate the adapter and index sequences necessary for NGS. (**C**) In tag-PCR, 50 ng of genomic DNA is required. Tagmentation is performed with in-solution transposomes to achieve both fragmentation and adapter ligation simultaneously. Two rounds of PCR amplification follow to amplify host–vector chimeric fragments and add the required NGS sequences for the Illumina platform. (**D**) In DIStinct-seq, up to 500 ng of genomic DNA is required per sample. Tagmentation is performed with bead-linked transposomes. Nested PCR is performed to specifically amplify host–vector chimeric fragments and attach adapter and index sequences. (**E**) After fragmenting the genomic DNA, each sequence is circularized through intra-molecular ligation using T4 ligase. Exonuclease treatment is then used to select only linearized DNAs. The circularized DNA, containing the target sequences, is cleaved and linearized by specific Cas9/sgRNA nucleases and then ligated with a sequencing adaptor. All methods rely on NGS to detect IS; they represent the junctions between the host and vector genomes. (Modified from https://BioRender.com/k63q271 (accessed on 18 January 2025)).

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
