# Peer review of "Leveraging Vector-Based Gene Disruptions to Enhance CAR T-Cell Effectiveness"

_cancers, 2025, doi:10.3390/cancers17030383_

Round 1
Reviewer 1 Report
Comments and Suggestions for Authors
I have read with great interest the review manuscript conducted by Oliveira et al. This is an overview of the impact of vector-based genomic disruptions during CAR T-cell manufacturing with a view to potentially exploit those disruptions which may lead to better CAR-T effectiveness. In addition, a variety of novel technologies which can be harnessed to detect the vector integration site, are presented. This topic is of great interest to the readers especially in the context of the FDA announcement regarding reported cases of secondary T-cell malignancies which raised safety concerns about potential vector-related genotoxicity, yet to be confirmed.
In general, the manuscript reads well and is relatively comprehensive given the limited clinical data. The quality of figures is satisfactory and most of the reference list is pertinent with the research area covering the relevant literature adequately. However, some revisions need to be undertaken. Please, see my comments below including both major and minor remarks:
1) Line 31: Please cite figure 1 later in the manuscript, maybe after a short sentence mentioning something basic regarding the manufacturing process. It is unusual citing a figure at the first line of a manuscript. In addition, the reader doesn’t really expect a figure about the manufacturing process in the context of this specific first sentence which refers to the general efficacy of CAR-T cells in haematological malignancies.
2) Line 33: Reference 3 not relevant, please replace. Cite some of the pivotal trials which led to the approval of commercial CAR-T products or the long-term follow-up of these trials.
3) Line 38-48: Although CAR-T structure is well known to the haematology readers, please, add citations.
4) Line 60, figure 1: “The T-cells are expanded, and stimulatory beads are removed before harvesting”. Authors maybe mean “before infusion” instead of “before harvesting”.
5) Line 61, figure 1: “After lymphodepleting chemotherapy, CAR T-cells are infused into the patient, either as a single dose or, as shown here, in three escalating doses over consecutive days or weeks”. Given that the readership of the Cancers journal is heterogeneous, I would clarify for the non-experts that split dosing is part of clinical trials as all commercial products are administered in a single dose, apart from the obecabtagene autoleucel product, recently approved by the FDA.
6) Line 75-77: I would also add manufacturing conditions, eg activation, as a factor having an impact on the number/quality of memory cells.
7) Line 84: “….we review how LVIS-based gene disruptions account for….”. I suggest retroviruses are also mentioned at this point of the manuscript along with a short part reporting main differences between retro- and lenti-viruses and potential advantages/disadvantages. This is essential given that the authors also mention retroviruses later throughout the manuscript and would also be interesting for the readership given there are approved commercial CAR-T products which are retrovirus-based. I understand that that the authors have a special interest and experience in lentivirus-based vectors, however, there is also a scientific reason why lentiviruses have dominated the gene therapy field.
8) Line 101, figure 2. Add explanation of the “UTR” abbreviation.
9) Line 126 and 129: Replace citation 39 with the original paper.
10) Line 142: Insert first author’s name followed by et all instead of number 33. Referencing the number is unusual in this context.
11) Line 147: Replace “survey” with the actual type of the study eg retrospective/prospective cohort etc. This type of study was not a survey.
12) Line 158: “Additionally, while some genes carried the vector insertion in multiple patients, the same site was never affected”. Was there any correlation of this finding with efficacy? Does this finding mean that the effect of the IS within a same gene is diverse and doesn’t translate to the same outcome across patients? Please, elaborate more on the clinical impact of this finding.
13) Line 163: “suggesting that some sites have a more prominent role in CD4 vs CD8 or even CD4-CD8- cells”. This is a speculation, there is no data so far supporting it. This may be true but the strong CD4 predominance could be also a natural functional evolution of both infused CD4/CD8 CAR T-subpopulations; for example, CD8 T-cells are significantly lower in numbers than CD4, CD8 T-cells decrease more drastically in older patients and CD8 T-cells have to comparable higher exhaustion than CD4 T-cells. In addition, CD4 T-cell cytotoxicity persists more in lymphopenic patients. Given this is a review and not an opinion manuscript, please, remove this sentence.
14) Line 166-169: These multiple disruptions may augment expansion and efficacy, but what about safety? The authors report that the impact depends on the precision of the IS but given that current viral vectors insert the payload in a semi-random fashion and there are not yet viral vectors developed which are extremely precise, reference to safety becomes inevitable.
15) Line 174: Insert first author’s name followed by et all instead of number 44.
16) Line 174-178: It is important to mention that these cases of ALL after gene therapy against X-SCID were developed with the use of previous generation viral vectors which were not that safe as the newer generation viruses.
17) Line 193-198: No need to explain CBL function in so many details, this is outside of the scope of this review. Generally, it is unclear why this case is presented is more details compared to the TET2 one. Please, develop both cases in a balanced way, either add more data for the TET2 or remove from the CBL.
18) Line 249: Insert first author’s name followed by et all instead of number 19.
19) Line 267: Insert first author’s name followed by et all instead of number 45.
20) Section 4 (line 272): This is a long paragraph. Consider in dividing in 2 paragraphs. Consider also transferring the first 2 sentences in section 2 as section 4 refers mainly to integration site analysis strategies.
21) Line 275: “….unpredictable outcomes….”. Please, clearly state what these outcomes involve.
22) Line 317: “….a relationship between the integrated sites and clonal behavior….”. Replace integrated sites with the abbreviation IS.
23) Section 5 (line 362): Apart from transposons, consider mentioning other technologies of genome or base editing which can achieve more precise site-specific integration, although even these technologies have been related with unintentional mutations of unknown significance. Consider also mentioning that research focusing on developing safer next-generation viral vectors may be another alternative solution, although there is still a lot of work to be done towards this direction.
24) Line 379-381: This applies only to insertions within coding regions of genes and not universally to all IS. Please, clarify in the text.
25) I understand that the main aim of this manuscript is to offer an overview of exploiting vector-integrated sites leading to better expansion and phenotypic memory CAR T-cells which may improve the clinical outcome. Although the authors have very briefly mentioned some safety aspects, the key message remains the beneficial impact of IS. Therefore, the manuscript is not presented in a balanced way to acknowledge safety and potential vector-related tumorigenesis. The readership would definitely expect to read a dedicated part on this aspect, especially after the FDA reported cases of secondary T-cell malignancies which is still unknown if they are vector mediated; however, it is a fact that after these reports a relevant box warning has been added to the SPC of all FDA approved products. Even when IS of next-generation safer viruses improves clonal CAR T-cell expansion and efficacy without explicitly leading to tumorigenesis, it is still unknown whether secondary hits eg additional acquired mutations or triggering effects such as inflammation mediated, could later lead to tumorigenesis. These are issues that should be addressed and equally developed either in a dedicated part or within the manuscript, and in the concluding section.
Author Response
-
Comments 1: Line 31: Please cite figure 1 later in the manuscript, maybe after a short sentence mentioning something basic regarding the manufacturing process. It is unusual citing a figure at the first line of a manuscript. In addition, the reader doesn’t really expect a figure about the manufacturing process in the context of this specific first sentence which refers to the general efficacy of CAR-T cells in haematological malignancies.
Response 1: We thank the reviewer for the comment. We added a brief description of the process and cited the figure later in the paragraph (lines 33-39).
Comments 2: Line 33: Reference 3 not relevant, please replace. Cite some of the pivotal trials which led to the approval of commercial CAR-T products or the long-term follow-up of these trials.Response 2: We have removed the reference and added other publications that are more relevant.
Comments 3: Line 38-48: Although CAR-T structure is well known to the haematology readers, please, add citations.
Response 3: We have added references as requested.
Comments 4: Line 60, figure 1: “The T-cells are expanded, and stimulatory beads are removed before harvesting”. Authors maybe mean “before infusion” instead of “before harvesting”.
Response 4: We thank the reviewer for the suggestion. We have modified the sentence.
Comments 5: Line 61, figure 1: “After lymphodepleting chemotherapy, CAR T-cells are infused into the patient, either as a single dose or, as shown here, in three escalating doses over consecutive days or weeks”. Given that the readership of the Cancers journal is heterogeneous, I would clarify for the non-experts that split dosing is part of clinical trials as all commercial products are administered in a single dose, apart from the obecabtagene autoleucel product, recently approved by the FDA.Response 5: We thank the reviewer for the comment. We added a sentence to clarify the treatment scheme in the figure legend as requested.
Comments 6: Line 75-77: I would also add manufacturing conditions, eg activation, as a factor having an impact on the number/quality of memory cells.
Response 6: We thank the reviewer for the comment. The sentence was modified to include manufacture conditions as a factor impacting the memory T cell repertoire.
Comments 7: Line 84: “….we review how LVIS-based gene disruptions account for….”. I suggest retroviruses are also mentioned at this point of the manuscript along with a short part reporting main differences between retro- and lenti-viruses and potential advantages/disadvantages. This is essential given that the authors also mention retroviruses later throughout the manuscript and would also be interesting for the readership given there are approved commercial CAR-T products which are retrovirus-based. I understand that that the authors have a special interest and experience in lentivirus-based vectors, however, there is also a scientific reason why lentiviruses have dominated the gene therapy field.
Response 7: We thank the reviewer for the comment. We added a short section in the introduction comparing retroviruses and lentiviruses (lines 31 to 60).
Comments 8: Line 101, figure 2. Add explanation of the “UTR” abbreviation.
Response 8: The explanation of UTR was added.
Comments 9: Line 126 and 129: Replace citation 39 with the original paper.
Response 9: We have added the original paper (lines 157 and 160).
Comments 10: Line 142: Insert first author’s name followed by et all instead of number 33. Referencing the number is unusual in this context.
Response 10: We modified the text to contain the author’s name.
Comments 11: Line 147: Replace “survey” with the actual type of the study eg retrospective/prospective cohort etc. This type of study was not a survey.
Response 11: We rephrased the sentence to avoid any confusion in interpreting what was the type of study.
Comments 12: Line 158: “Additionally, while some genes carried the vector insertion in multiple patients, the same site was never affected”. Was there any correlation of this finding with efficacy? Does this finding mean that the effect of the IS within a same gene is diverse and doesn’t translate to the same outcome across patients? Please, elaborate more on the clinical impact of this finding.
Response 12: The reviewer brings up a good point. We hypothesize that depending on the specific region wherein the vector integrates into the gene, the response can be variable, however, more studies are needed to confirm this hypothesis, and our group is currently working on it.
As an example, in two previous reports where TET2 (reference 49) and CBL (reference 52) were disrupted in patients treated with CAR T-cells, their specific affected sites/alleles were correlated with positive outcomes. Especially in the case of TET2, several mechanistic studies performed later by the Sadelain group and others have shown that the complete ablation of this gene can lead to clonal malignancy in different cancer models, but this was not the case in patient 10, where the gene was still partially functional (one allele disruption was caused by vector integration and a hypomorphic mutation was observed in the second allele) and this patient is still in remission until the present day, 13 years later. This highlights the importance of investigating how certain gene disruptions affect T cell biology and how specific mutations in those genes impact the outcome of these therapies long-term.
We made a comment to elaborate the sentence (lines 190-193).
Comments 13: Line 163: “suggesting that some sites have a more prominent role in CD4 vs CD8 or even CD4-CD8- cells”. This is a speculation, there is no data so far supporting it. This may be true but the strong CD4 predominance could be also a natural functional evolution of both infused CD4/CD8 CAR T-subpopulations; for example, CD8 T-cells are significantly lower in numbers than CD4, CD8 T-cells decrease more drastically in older patients and CD8 T-cells have to comparable higher exhaustion than CD4 T-cells. In addition, CD4 T-cell cytotoxicity persists more in lymphopenic patients. Given this is a review and not an opinion manuscript, please, remove this sentence.
Response 13: We appreciate the reviewer’s comment. We removed the sentence.
Comments 14: Line 166-169: These multiple disruptions may augment expansion and efficacy, but what about safety? The authors report that the impact depends on the precision of the IS but given that current viral vectors insert the payload in a semi-random fashion and there are not yet viral vectors developed which are extremely precise, reference to safety becomes inevitable.
Response 14: We thank the reviewer for the comment. We added a sentence about safety on line 203.
Comments 15: Line 174: Insert first author’s name followed by et all instead of number 44.
Response 15: We have added the author’s name as suggested (line 208).
Comments 16: Line 174-178: It is important to mention that these cases of ALL after gene therapy against X-SCID were developed with the use of previous generation viral vectors which were not that safe as the newer generation viruses.
Response 16: We thank the reviewer for pointing this out. We have added a sentence to clarify the type of viral vector used (lines 212-214).
Comments 17: Line 193-198: No need to explain CBL function in so many details, this is outside of the scope of this review. Generally, it is unclear why this case is presented is more details compared to the TET2 one. Please, develop both cases in a balanced way, either add more data for the TET2 or remove from the CBL.
Response 17: We thank the reviewer for the comment. We removed the detailed information about CBL.
Comments 18: Line 249: Insert first author’s name followed by et all instead of number 19.
Response 18: We have added the author’s name as requested (line 295).
Comments 19: Line 267: Insert first author’s name followed by et all instead of number 45.
Response 19: We have added the author’s name as requested (line 313).
Comments 20: Section 4 (line 272): This is a long paragraph. Consider in dividing in 2 paragraphs. Consider also transferring the first 2 sentences in section 2 as section 4 refers mainly to integration site analysis strategies.
Response 20: We appreciate the comment. We have divided the paragraph in 3.
Comments 21: Line 275: “….unpredictable outcomes….”. Please, clearly state what these outcomes involve.
Response 21: We have added examples on lines 321-322,
Comments 22: Line 317: “….a relationship between the integrated sites and clonal behavior….”. Replace integrated sites with the abbreviation IS.
Response 22: We have replaced the term with the abbreviation.
Comments 23: Section 5 (line 362): Apart from transposons, consider mentioning other technologies of genome or base editing which can achieve more precise site-specific integration, although even these technologies have been related with unintentional mutations of unknown significance. Consider also mentioning that research focusing on developing safer next-generation viral vectors may be another alternative solution, although there is still a lot of work to be done towards this direction.
Response 23: We thank the reviewer for the comment. We have included more technologies of CAR delivery (lines 422-439).
Comments 24: Line 379-381: This applies only to insertions within coding regions of genes and not universally to all IS. Please, clarify in the text.
Response 24: We added a sentence on line 451.
Comments 25: I understand that the main aim of this manuscript is to offer an overview of exploiting vector-integrated sites leading to better expansion and phenotypic memory CAR T-cells which may improve the clinical outcome. Although the authors have very briefly mentioned some safety aspects, the key message remains the beneficial impact of IS. Therefore, the manuscript is not presented in a balanced way to acknowledge safety and potential vector-related tumorigenesis. The readership would definitely expect to read a dedicated part on this aspect, especially after the FDA reported cases of secondary T-cell malignancies which is still unknown if they are vector mediated; however, it is a fact that after these reports a relevant box warning has been added to the SPC of all FDA approved products. Even when IS of next-generation safer viruses improves clonal CAR T-cell expansion and efficacy without explicitly leading to tumorigenesis, it is still unknown whether secondary hits eg additional acquired mutations or triggering effects such as inflammation mediated, could later lead to tumorigenesis. These are issues that should be addressed and equally developed either in a dedicated part or within the manuscript, and in the concluding section.
Response 25: We appreciate the comment, and we agree that this aspect should be incorporated in our manuscript. We have discussed the concerns raised by the FDA and the cases reporting secondary malignancies followed by CAR T-cell therapy (lines 455-466).
4. Response to Comments on the Quality of English Language
Point 1: Quality of English Language
(x) The quality of English does not limit my understanding of the research.
( ) The English could be improved to more clearly express the research.Response 1: We thank the reviewer for the comment.
5. Additional clarifications
N/A
Reviewer 2 Report
Comments and Suggestions for Authors
This is a very nice, well written review focused on a small but very relevant topic in CAR-T cells therapies. In general, the manuscript would benefit from figures with higher resolution and a double-checked reference list (especially ref. 35 to 40) as well as a bigger overview of literature assessing safety issues in CAR T cell therapy. Also, the interesting targets (TET2, JAK3, Cbl-b) can be exploited in more detail to show benefits and disadvantages of integration into these genes. This would also better align with the chosen title. Additionally, numbers up to twelve should be written out and there should be consistently used T cells or T-cells. More considerations to increase the impact of the manuscript are suggested below.
- Line 40: remove “so-called”, scFv is commonly used and known
- Line 42: remove “cells” after “target”, CAR can bind also to on-target of-tumor cells which are not a target cell
- Line 44: use “part” instead of “portion”
- Line 46: introduce “signal 1” before mentioning it. Subsequently, signal 2 should also be explained
- Line 49: “derived from” should be placed before CD28
- Line 49: benefits of CD28 based CARs should be explained
- Line 177: rephrase sentence
- Line 232: too many gaps before ref 19
- Line 303: explain tagmentation technique
- Line 370: rephrase sentence
- Line 373: include genetic modified CAR T cell products
- Line 375: this sentence is very hard to read, please modify
- Discussion: give overview over reported cases of malignant transformation and offer solutions for risk of targeting oncogenes
- References which should be incorporated:
o https://www.nejm.org/doi/full/10.1056/NEJMp2400209
o https://ashpublications.org/blood/article/142/Supplement%201/6939/504399/CAR-T-Cell-Lymphoma-Post-Ciltacabtagene-Autoleucel
o https://www.nature.com/articles/s41591-024-02826-w
o https://pubmed.ncbi.nlm.nih.gov/33462140/
Author Response
|
Comments 1: Line 40: remove “so-called”, scFv is commonly used and known Response 1: We thank the reviewer for the suggestion. We have removed the term. Comments 2: Line 42: remove “cells” after “target”, CAR can bind also to on-target of-tumor cells which are not a target cell Response 2: The word was removed. Comments 3: Line 44: use “part” instead of “portion” Response 3: We changed the wording in line 66. Comments 4: Line 46: introduce “signal 1” before mentioning it. Subsequently, signal 2 should also be explained Response 4: We have added the explanation of signal 1on line 68. Comments 5: Line 49: “derived from” should be placed before CD28 Response 5: The change was made on line 71. Comments 6: Line 49: benefits of CD28 based CARs should be explained Response 6: We appreciate the reviewer’s comment. We acknowledge the benefits and potency of the CD28-based CARs; however, our focus was not to compare the different constructs but rather give a brief overview of the elements contained in them. Comments 7: Line 177: rephrase sentence Response 7: We were unsure of where exactly the sentence should be rephrased, however lines 212-216 were modified based on both reviewers’ comments. Comments 8: Line 232: too many gaps before ref 19 Response 8: We have removed a few words and referenced the study earlier in the paragraph. Comments 9: Line 303: explain tagmentation technique Response 9: We have added the explanation to the technique. Comment 10: Line 370: rephrase sentence Response 10: We have rephrased the sentence (line 451). Comments 11: Line 373: include genetic modified CAR T cell products Response 11: We have included other approaches on lines 432-449 based on both reviewers’ comments. Comments 12: Line 375: this sentence is very hard to read, please modify Response 12: We have modified the sentence and made it clearer. Comments 13: Discussion: give overview over reported cases of malignant transformation and offer solutions for risk of targeting oncogenes References which should be incorporated: o https://www.nejm.org/doi/full/10.1056/NEJMp2400209 o https://ashpublications.org/blood/article/142/Supplement%201/6939/504399/CAR-T-Cell-Lymphoma-Post-Ciltacabtagene-Autoleucel o https://www.nature.com/articles/s41591-024-02826-w o https://pubmed.ncbi.nlm.nih.gov/33462140/ Response 13: We thank the reviewer for the comments. We have added a dedicated section to the cases that were reported and those being investigated by the FDA (lines 455-466). 4. Response to Comments on the Quality of English Language |
|
Point 1: Quality of English Language (x) The quality of English does not limit my understanding of the research.
|
|
Response 1: We thank the reviewer for the comment.
|
|
|
|
|
Round 2
Reviewer 1 Report
Comments and Suggestions for Authors
Thank you for addressing all comments. I noted in the revised manuscript that the sentence "suggesting that some sites have a more prominent role in CD4 vs CD8 or even CD4-CD8- cells" on line 185-186 has not been removed. Please, remove as previously agreed.
After this correction, I don't have further comments.
Author Response
|
Comments 1: Thank you for addressing all comments. I noted in the revised manuscript that the sentence "suggesting that some sites have a more prominent role in CD4 vs CD8 or even CD4-CD8- cells" on line 185-186 has not been removed. Please, remove as previously agreed.
After this correction, I don't have further comments.
|
|
|
Reviewer 2 Report
Comments and Suggestions for Authors
The authors adressed all points and updated the figures.
Author Response
|
Comments 1: The authors addressed all points and updated the figures. |
|
|